# Docosahexaenoic Acid Ameliorates Contextual Fear Memory Deficits in the Tg2576 Alzheimer’s Disease Mouse Model: Cellular and Molecular Correlates

**DOI:** 10.3390/pharmaceutics15010082

**Published:** 2022-12-27

**Authors:** Sara Badesso, Paz Cartas-Cejudo, Maria Espelosin, Enrique Santamaria, Mar Cuadrado-Tejedor, Ana Garcia-Osta

**Affiliations:** 1Neurosciences Program, Center for Applied Medical Research (CIMA), University of Navarra, IdiSNA, 31008 Pamplona, Spain; 2Clinical Neuroproteomics Unit, Navarrabiomed, Hospital Universitario de Navarra (HUN), Universidad Pública de Navarra (UPNA), IdiSNA, 31008 Pamplona, Spain; 3Department of Pathology, Anatomy and Physiology, School of Medicine, University of Navarra, IdiSNA, 31008 Pamplona, Spain

**Keywords:** Alzheimer’s disease, DHA, synapse

## Abstract

Docosahexaenoic acid (DHA), the most abundant polyunsaturated fatty acid in the brain, is essential for successful aging. In fact, epidemiological studies have demonstrated that increased intake of DHA might lower the risk for developing Alzheimer’s disease (AD). These observations are supported by studies in animal models showing that DHA reduces synaptic pathology and memory deficits. Different mechanisms to explain these beneficial effects have been proposed; however, the molecular pathways involved are still unknown. In this study, to unravel the main underlying molecular mechanisms activated upon DHA treatment, the effect of a high dose of DHA on cognitive function and AD pathology was analyzed in aged Tg2576 mice and their wild-type littermates. Transcriptomic analysis of mice hippocampi using RNA sequencing was subsequently performed. Our results revealed that, through an amyloid-independent mechanism, DHA enhanced memory function and increased synapse formation only in the Tg2576 mice. Likewise, the IPA analysis demonstrated that essential neuronal functions related to synaptogenesis, neuritogenesis, the branching of neurites, the density of dendritic spines and the outgrowth of axons were upregulated upon-DHA treatment in Tg2576 mice. Our results suggest that memory function in APP mice is influenced by DHA intake; therefore, a high dose of daily DHA should be tested as a dietary supplement for AD dementia prevention.

## 1. Introduction

Clinical and epidemiological studies have shown that supplementing the diet with omega-3 fatty acids (FAs) reduced the risk of dementia development [1]. Omega-3 FAs are indispensable components of neuronal membranes and are essential for membrane fluidity, neurotransmitter release, signal transduction and neuroinflammation processes [2]. Docosahexaenoic acid (DHA, 22:6 n-3) is the most abundant polyunsaturated FA in the mammalian brain, representing ~30% of the lipids in the human brain [3], with very high levels in neuronal membranes, synaptic terminals and the myelin sheath [4]. It is essential to maintain optimal DHA levels during the development and maturation of the brain and it is also critical for successful aging of the adult brain [5,6].

A decrease in serum DHA levels is associated with memory decline [7], more cerebral amyloidosis and brain atrophy [8,9]; therefore, decreased DHA levels have been found in the brains of people affected with AD, specifically in memory-related areas such as the hippocampus [10,11,12,13,14,15,16].

These observations motivated the implementation of several clinical trials to evaluate the effect of DHA (alone or in combination with other omega-3 FAs) in patients with mild to moderate AD [16,17,18,19,20]. Overall, to date, none of the studies have shown a clear positive effect of DHA or its combinations on cognitive function or brain atrophy. This could be because DHA is not entering the central nervous system (CNS) in a sufficient concentration to change the course of the disease. Accordingly, a recent study using a treatment of 2 g DHA per day for 26 weeks, found that the levels of DHA in the CSF increased by only 28% (whereas the increase in plasma was 200%), with no differences in cognitive function or brain volume compared to placebo. This study suggested that previous studies using much lower DHA doses may represent negative studies due, at least in part, to its scarce bioavailability in the brain, particularly in APOE4 carriers [17]. 

The effect of long-term administration of omega-3 FAs has been also studied in different AD experimental models [21,22,23,24,25,26], although the results are quite heterogeneous, probably due to the large variation in animal models, doses and duration of treatments [27]. Furthermore, in most cases, the dose of DHA was given as a supplement to a diet containing other FAs, which makes it difficult to assess the individual effect of the specific dose of DHA received. Despite the variability, the overall conclusions have been that long-term omega-3 FA supplementation increases DHA levels in the brain and ameliorates AD pathology. 

Different mechanisms have been proposed to explain the beneficial effect of DHA in cognition, including its anti-oxidative, anti-inflammatory [28] and anti-apoptotic properties, the facilitation of transport of glucose into the brain [29] and the improvement of vascular health [30]. Therefore, the aim of the current study was to analyze the effect of a high-dose (450 mg/kg) concentrated DHA-triglyceride formula provided by Nua Biological Innovations SL (Vizcaya, Spain), in comparison with a vehicle (sesame oil), on cognitive function and AD pathology in aged Tg2576 mice, an amyloid over-expression model of AD. This formulation complied with a very rigorous quality standard (five-star IFOS program), which guaranteed that the oil was not oxidized and was practically free of heavy metals (mercury, cadmium, lead and arsenic), polychlorinated biphenyls, furans or dioxins [31]. Furthermore, taking into account that most of the studies on the effects of DHA in AD models have been focused on changes in amyloid burden and in the synaptic pathology, our aim was to unravel the further underlying molecular mechanism of a possible transgene-dependent effect of DHA on synaptogenesis and in memory formation, by using RNA sequencing.

## 2. Results

### 2.1. Fear Memory and Spatial Memory Performance of 18-Month-Old Tg2576 Mice Chronically Treated with DHA

To evaluate the effect of DHA on memory function, we used the fear conditioning and the Morris water maze tasks. In the fear cognition task, a two-way ANOVA analyzed the freezing time, revealing a significant effect of genotype [F(1,44) = 35.43, *p* < 0.0001], without a significant effect of treatment [F(1,44) = 2.80, *p* = 0.10], and with significant interaction of the two factors [F(1,45) = 6.84, *p* = 0.01]. A Tukey’s post hoc test confirmed that Tg2576 mice exhibited significantly less freezing time compared with WT mice (*p* < 0.0001) in the test session, and DHA intake ameliorated the memory deficits observed in Tg2576 mice (*p* < 0.05). However, no significant effect was observed in non-Tg mice after DHA treatment (Figure 1A).

Afterwards, spatial memory was also assessed in the MWM. For each group, the escape latency over time was analyzed using the non-parametric Friedman test. Latency to reach the platform decreased in every group as the training sessions progressed, except in the Tg2576-DHA group (WT-Veh: *p* < 0.0001; WT-DHA: *p* < 0.001; Tg2576-Veh: *p* < 0.05; Tg2576-DHA: *p* = 0.09), indicating that all animals were able to learn the platform location, except for the Tg2576-DHA group (Figure 1B). Next, comparisons among the groups were assessed using a two-way ANOVA and Tukey’s post hoc test for each day. Two-way ANOVA revealed a significant effect of genotype on every day of the hidden platform (day 1: F(1,33) = 9.22, *p* < 0.01; day 2: F(1,33) = 6.31, *p* < 0.05; day 3: F(1,33) = 9.37, *p* < 0.01; day 4: F(1,33) = 17.90, *p* < 0.001; day 5: F(1,33) = 14.13, *p* < 0.001), with no significant effect of the treatment, nor the interaction (Figure 1B). 

Exploration time analysis in the target quadrant, using two-way ANOVA, showed a significant main effect of genotype (F(1,33) = 9.96, *p* < 0.01) but no significant main effect of treatment, nor significant interaction (Figure 1C). This result indicates that a 5-week treatment with a high dose of DHA ameliorated freezing memory deficits, but was not sufficient to restore spatial memory and retention deficits observed in aged Tg2576 mice. 

### 2.2. Synaptic Pathology in the Brain of 18-Month-Old Tg2576 Mice Chronically Treated with DHA

Different authors have suggested that DHA promotes synaptogenesis in different animal models [21,32]. Thus, we measured the dendritic spine density of the mice hippocampi after Golgi–Cox staining. Two-way ANOVA analysis revealed a significant effect of treatment (F(1,428) = 32.73 *p* < 0.0001) and significant main interaction of genotype × treatment (F(1,428) = 6.97 *p* < 0.01). As is shown in Figure 1D, the spine density in CA1 pyramidal neurons was significantly higher in the Tg2576-DHA than in Tg2576-Veh group (Tg2576-DHA, 2.52 ± 0.7/μm; Tg2576-Veh, 2.04 ± 0.5/μm; *p* < 0.0001). Although a tendency was observed in the WT-DHA group compared with the WT-Veh group, the increase was not significant (WT-DHA, 2.42 ± 0.5/ μm; WT-Veh, 2.24 ± 0.5/ μm; *p* = 0.13). These data confirmed previous evidence showing that DHA promotes dendritic spine formation in APP mice [21]. Our data indicate that DHA is sufficient to induce a cognitive enhancement and synapse formation in Tg2576 mice, but not in control Tg(-) (or WT) mice, suggesting that APP mice have an added susceptibility to DHA levels in the diet and/or in the brain.

### 2.3. Amyloid Pathology and Neuroinflammatory Markers in the Brain of 18-Month-Old Tg2576 Mice Chronically Treated with DHA

To explore a possible effect of DHA treatment on amyloid pathology in Tg2576 mice brains, ELISA was used to determine the concentration of Aβ42 soluble in a 2% SDS buffer (detergent soluble aggregates) and guanidinium-chloride-treated (insoluble aggregates, mostly plaques) parietotemporal cortex extracts. No significant differences in Aβ content in any of the different extracts were detected between the transgenic mice groups, irrespective of whether they received DHA or not (Figure 2A). Accordingly, we analyzed APP processing and no differences were observed in the ratio of the APP-derived fragments C99/C83, indicating that DHA treatment does not affect the production of the proteolytic products C83 and C99. Considering that previous studies have reported changes in amyloid pathology after DHA treatment, we wanted to confirm that this was not the case in this study. Thus, as an indirect parameter, and considering that amyloid plaques are surrounded by reactive astrocytes in APP mice, we next analyzed the expression of brain GFAP using a Western blot analysis. As is depicted in Figure 2B, a two-way ANOVA analysis again revealed a marked significant effect of genotype (F(1,33)= 99.65, *p* < 0.0001), but no significant effect of the treatment nor interaction. 

Next, we explored whether other microglial markers, such as the microglia-associated triggering receptor expressed on myeloid cells (TREM2) and CD68, that may be altered in the presence of amyloid pathology, were affected by DHA treatment. For both genes, a two-way ANOVA analysis showed a main effect of genotype (F(1,11) = 14.70, *p* < 0.01 for TREM2 and F(1,11) = 12.92, *p* < 0.01 for CD68), indicating that TREM2 and CD68 were significantly up-regulated in Tg2576 transgenic mice compared with non-transgenic mice. Additionally, in both cases a main effect of treatment was detected (F(1,11) = 17.78, *p* < 0.01 for TREM2 and F(1,11) = 6.67, *p* < 0.05 for CD68). Tukey’s multiple comparison tests revealed that DHA significantly reduced the expression of TREM2 (*p* < 0.01) and CD68 (*p* < 0.05) in Tg2576 mice, to levels comparable to that of wild-type mice (Figure 2C). Regarding the levels of pro-inflammatory markers, IL1β and TNFα, a two-way ANOVA showed a main effect of genotype (F(1,23) = 9.73, *p* < 0.01 for IL1β and F(1,23) = 11.28, *p* < 0.01 for TNFα); however, a main effect of treatment, although no interaction, was also found for TNFα (F(1,23) = 4.24, *p* < 0.05). Tukey’s multiple comparison tests revealed significant differences in TNFα expression in the Tg2576 vehicle compared to both groups of wild-type mice (Figure 2D). 

### 2.4. RNA-Seq Analysis

To explore the molecular mechanism underlying this transgene-dependent effect of DHA on synaptogenesis and in memory formation, we performed a transcriptomic analysis of the mice hippocampi (GEO accession number GSE217430). Among the DHA-regulated genes, only 6 were similarly modified in both genotypes, WT and Tg2576 mice (Figure 3A), while 102 genes were differentially regulated between WT and Tg2576 mice, thus indicating that an important transgene-dependent effect was contributing to the action of DHA (Appendix A). Among the similarly modified genes upon DHA treatment, Sec14l5 (SEC14 like lipid binding 5) and SSPO (subcommissural organ-spondin) genes were downregulated, while Nr4a2 (nuclear subfamily receptor 4 member 2), Cbln4 (cerebellin 4 precursor), Lefty2 (left–right determination factor 2) and Prss8 (serine protease 8) were upregulated (Figure 3A).

Regarding downregulated genes, while no studies have linked Sec14l5 with synaptogenesis, SESTD1, a phospholipid-binding protein containing a lipid-binding SEC14-like domain, seems to negatively regulate dendritic spine density in cultured hippocampal neurons [33]. Nevertheless, SSPO, which is a multidomain protein of the extracellular matrix, has already been involved in neuronal survival and neurite extension [34]. Among upregulated genes upon DHA treatment, two were of special interest. On the one hand, Cbln4 is a member of a small family of secreted synaptic proteins that seems to play an essential role in the formation and maintenance of inhibitory GABAergic connections. Interestingly, Chacón et al. stated that Cbln4 expression was significantly decreased in the hippocampus of a mouse model of AD and that its overexpression in cultured hippocampal neurons rescued neurons from Aß-induced death, which suggests a therapeutic potential for Cbln4 in AD [35]. On the other hand, recent studies have also pointed out a role for the transcription factor Nr4a2 in hippocampal synaptic plasticity and cognitive function [36]. It seems that by regulating the neural networks required for memory formation and consolidation, transcription factors act as pivotal players underlying synaptic plasticity. Taking this into consideration, Nr4a2 was next validated by quantitative real-time PCR (qRT-PCR) (Figure 3B). 

A two-way ANOVA showed a main effect of treatment (F(1,39) = 9.45, *p* < 0.01), but not for genotype nor interaction. Tukey’s multiple comparison tests revealed that Nr4a2 was significantly increased in Tg2576-DHA-treated mice when compared to WT (*p* < 0.05) or Tg2576 vehicle-treated (*p* < 0.05) mice, suggesting that this transcription factor could be a central key player, through which DHA may regulate gene expression profiles upstream of certain synaptic therapeutic targets. 

### 2.5. Predictive Activation Profile of Pathways and Neuronal Functions upon DHA Treatment

Subsequently, based on transcriptomic datasets, we performed a systems biology analysis using Ingenuity Pathway Analysis (IPA) software to identify differentially regulated canonical pathways between vehicle-treated mice (Tg2576 vs. WT) and DHA-treated mice (Tg2576 vs. WT) (Figure 4A). Interestingly, the synaptogenesis signaling pathway, CREB signaling in neurons and calcium signaling pathways were among the inhibited pathways in the Tg2576 vs. WT (vehicle) mice, and they were partially rescued upon DHA treatment (Figure 4A,B). The focal adhesion kinase (FAK) signaling pathway, which has been described as regulating neuronal growth, synaptic plasticity and hippocampus-dependent learning and memory [37], was one of the signaling routes that tended to be activated upon DHA treatment (Figure 4A,B). Likewise, STAT3 and G alpha 12/13 signaling pathways, which have been reported to be involved in synaptic plasticity [38], and in neuronal migration, axonal guidance and neurotransmitter release [39], respectively, were also among the significant predictive inhibited pathways in the Tg2576 mice compared to WT mice (untreated), and they became activated upon DHA treatment (Figure 4A,C). Changes in the morphology and density of dendritic spines are believed to be crucial for maintaining synaptic function and plasticity. Thus, these predictions, based on differential transcriptomic fingerprints, together with behavioral data and with the dendritic spine density analysis (Figure 1), indicate that DHA treatment is able to partially ameliorate fear memory deficits of Tg2576 mice through the reactivation of these pathways. Nonetheless, in WT animals with a preserved memory function, no enhancement of memory or significant increase in spine density was appreciated.

Similar results were obtained when the predictive analysis was focused on neuronal function (Figure 5A), since long-term potentiation, neuritogenesis, the branching of neurites, the density of dendritic spines or the outgrowth of axons were among the activated biofunctions with DHA treatment (Figure 5A,B). The activation of these biofunctions mainly support the regulation of spine density observed at the CA1 pyramidal neurons of the hippocampus of DHA-treated Tg2576 mice. These results, as many authors have already suggested, reinforce the concept that DHA could exert its beneficial effects on AD-mice via neuroplasticity enhancement [40].

Strikingly, other specific functions that seemed to be associated with DHA treatment were “cell viability of neurons” and “apoptosis of neurons”, suggesting a plausible neuroprotective role of DHA (Figure 5A,B).

### 2.6. Mechanistic Hypothesis of DHA Based on Differential Transcriptomic Profiling between Vehicle-Treated Mice (Tg2576 vs. WT) and DHA-Treated Mice (Tg2576 vs. WT)

Based on differential transcriptomic profiling obtained from vehicle-treated (Tg2576 vs. WT) (Figure 6A) and DHA-treated (Tg2576 vs. WT) mice, together with the IPA knowledgebase, a mechanistic hypothesis for DHA was generated. Functional networks using a precomputed table containing inferred relationships between molecules, functions, diseases and pathways were used to score them with a machine learning algorithm operating on prior fundamental knowledge. The most significantly activated (in orange; positive Z-score) or inhibited (in blue; negative z-score) upstream regulators, diseases, functions and pathways highly related to the differential RNA-seq datasets were used to create the mechanistic networks (Figure 6). 

Intriguingly, and in accordance with previous analysis (Figure 4 and Figure 5), important pathways related to synaptic plasticity, such as branching of neurites, long-term potentiation, and quantity of synaptic vesicles that were significantly inhibited in Tg2576 vehicle-treated mice when compared with WT vehicle-treated mice (Figure 6A), did not significantly change between Tg2576 and WT DHA-treated mice. These results support the idea that DHA may trigger the induction of neurite development, synaptogenesis and expression of plasticity-related genes, and that it could be considered as a potential therapeutic compound to prevent and/or even slow down AD progression. 

The mechanistic hypothesis generated also indicated that the beneficial effect of DHA may also be mediated by its action through inflammatory markers. Specifically, it showed that the activation of TREM signalling in Tg vs. WT mice (Figure 6A) was alleviated with DHA (Figure 6B), while the activation of neuroinflammatory markers such as ILβ and TNFα (Figure 6A) was not observed upon DHA treatment (Figure 6B). It should be noted that these results correlate with the analysis of RT-qPCR neuroinflammatory markers (Figure 2), thus supporting the proposed mechanistic hypothesis. 

## 3. Discussion

DHA is the main structural n-3 fatty acid in the brain and is essential for a functional nervous system. The benefits of DHA intake on cognitive function are widely recognized. Notwithstanding, the positive effect seems more convincing in healthy adults with a mild memory complaint than in diagnosed AD patients, rather indicating the role of prevention [2,41]. Even so, some studies fail to demonstrate a protective effect of fatty acid intake against the development of mild cognitive impairment [42,43]. An interaction between the stage of the disease and the ApoE genotype may explain the inconsistent results obtained with DHA supplementation. In fact, it has been suggested that APOE4 carriers showed a higher peripheral DHA catabolism that can limit DHA availability to the brain [9,44,45,46]. On the other hand, where long-term DHA supplementation may slow cognitive decline in an early prodromal stage of dementia, it would not affect memory decline in an advanced stage of AD [47]. 

Similar beneficial effects for DHA on different AD models have been described [2,23,27]. However, the evidence is still far from convincing, since many other studies have failed to corroborate those results [48,49]. The disagreement in the outcomes may be due to disparity in the duration and dosage of the treatment and/or the different animal models used. It is important to highlight that the greatest benefits have been observed when treatment begins before the onset of symptoms. In addition, similarly to humans, it has been found that APOE4 mice showed a lower brain DHA uptake than APOE2 and APOE3 mice [50].

As a result, it can be said that further studies that are also focused on elucidating the molecular mechanisms underlying the potential beneficial effect of DHA on memory function are required. Considering that, in most of the studies carried out with rodents, DHA was administered as a supplement in the water or pellets, without controlling the amount of DHA intake in the diet, we decided to perform a more rigorous study using a controlled high dose of DHA and an AD mouse model with a well-established phenotype. Furthermore, in order to gain insight into the main pathways involved in the beneficial effect of DHA, an RNA-seq analysis of the hippocampi of DHA-treated mice vs. vehicle-treated mice (Tg2576 and WT mice) was undertaken. Our results confirmed a slight, but significant protective effect of DHA, that seems to be mediated by an enhancement of the formation of synaptic contacts. However, our treatment regimen was not capable of reducing the amyloid pathology or the neuroinflammation that accompanies this condition. This is partially consistent with the study by Hooijmans et al., that found a decrease in the amount of vascular Aβ but no changes in brain parenchymal Aβ accumulation in APP/PS1 mice receiving a DHA-enriched diet [30].

In contrast, other studies reported a significant reduction in amyloid deposition after DHA supplementation. Specifically, in a study similar to this one, in which DHA was administered over 5 months to Tg2576 mice, DHA was able to reduce Aβ_42_ accumulation by decreasing APP processing [24]. Accordingly, similar results were reported when DHA was administered to APP/PS1 [25,26,51] or 3xTg [52] mice; but again, in each of these studies, the treatment regimen was much longer (from at least 4 to 12 months). Our results indicate that a 5-week DHA treatment ameliorates fear memory deficits in aged Tg2576 mice; however, the duration of the treatment was probably not long enough to have an impact on amyloid pathology. Thus, taking into consideration that no changes in amyloid were observed, but an enhancement of memory function and an increase in dendritic spines were appreciated upon DHA treatment, we could suggest that: (i) DHA seems to restore synaptic pathology without affecting amyloid burden, and (ii) despite the important amyloid burden present in the brain of 18-month-old Tg2576 mice, memory function could be ameliorated. In any case, it is also important to indicate that this 5-week DHA treatment was not sufficient to restore the severe spatial memory deficits of aged Tg2576 mice (see MWM data in Figure 1B). Accordingly, previous studies using therapeutic interventions have also showed different outcomes in the contextual fear memory and MWM behavioral tests [53,54]. It should be noted that the hippocampus is a key region in navigation and spatial learning [55], but it is also essential for remembering aversive contexts [56]. However, these two behavioral tasks show varying sensitivity and, more importantly, seem to engage task-specific neurons with different key molecular players. Specifically, in our case, the increase in CA1 dendritic spine density upon DHA supplementation observed in aged-Tg2576 mice seem to underly fear memory restoration but not spatial memory impairment. In contrast with other studies [57], it may seem shocking that, although there is a trend, no significant differences in dendritic spine density were observed between WT and Tg2576 vehicle groups (*p* = 0.06, Figure 1D). Nonetheless, this is probably because there is an age-dependent regression of the spine density (mice were approximately 22 months old) that makes it lose significance when comparing the wild-type group with the transgenic group [58,59].

These conclusions were, in part confirmed by the RNA-seq analysis. On one hand, the analysis demonstrated that the Tg2576 model had a clear AD phenotype with memory loss, abundant amyloid pathology and synaptic loss compared to age-matched wild-type mice. On the other hand, the analysis showed that, unlike the Tg2576 vehicle group, the Tg2576 DHA-treated group did not show significant differences in synaptic pathology compared to wild-type mice receiving the same treatment. Neuroinflammation and/or APP-related pathways, however, seemed to be unaltered upon DHA treatment. Similarly, the bioinformatic predictive analysis of the transcriptomic datasets using IPA also demonstrated that, upon DHA treatment, activated pathways and neuronal functions related to synaptogenesis, neuritogenesis, the branching of neurites, the density of dendritic spines and the outgrowth of axons were activated, in addition to the activation of CREB and calcium signaling pathways. These results shed light on the molecular mechanisms by which DHA restored dendritic spines loss and improved memory in Tg2576 mice. In accordance with this, previous studies have already described that DHA supplementation significantly affects hippocampal neuronal development and synaptic function by inducing neurite growth, synaptogenesis, synapsin and glutamate receptor expression [60,61]. In the same way, its deprivation during development seems to decrease synapsins and glutamate receptor subunits and alter long-term potentiation in 18-day-old pups [61]. It is important to highlight that our results are in accordance with previous studies reporting that DHA is highly enriched in synapses and that it plays an important role in the expression of many synaptic essential proteins for the integrity of excitatory synapses such as drebrin and postsynaptic density protein 95 (PSD-95) [21,62]. Moreover, it should be noted that the activation of the BDNF pathway, which was also observed to be activated in our IPA analysis upon DHA treatment, had been previously observed to be modulated by DHA after brain trauma [63]. In this sense, our study serves to molecularly confirm and validate some of the beneficial effects already observed with DHA and to point towards specific synaptic pathways underlying its beneficial effect.

Taking all these data together and considering that, in prodromal AD patients, synapse loss is the best correlate of cognitive impairment, we can conclude that DHA is one of the best dietary supplements to prevent synaptic deficits and/or strengthen synaptic plasticity. In this regard, is important to consider that, in humans, synapse loss precedes neuronal death and represents the prodromal phase of the disease, which might be represented by AD mice with an established AD phenotype, such as those in our study (18-month-old Tg2576 mice). Accordingly, it seems that the beneficial effects of DHA, as already mentioned, might not be sufficient to restore memory deficits in people with an already established pathology; however, they may be effective in preventing or delaying disease progression towards dementia. This has previously been suggested, after several epidemiological studies demonstrated an association between DHA intake and/or DHA blood levels with lower AD risk [64]. In fact, clinical trials using omega-3 fatty acid supplements for preventing dementia have already been undertaken. However, to date, no positive outcomes on cognition have been demonstrated, probably because low doses were used (lower than 1 g) [65,66]. More recently, as has already been mentioned, different studies have suggested that larger doses of DHA are probably required for adequate brain bioavailability, particularly in APOE4 carriers, in whom plasma DHA metabolism is altered, affecting the transport of DHA into the brain [9,17]. In this regard, authors of one such study are currently testing the effect of high-dose (2 g per day) DHA supplementation on (i) CSF fatty acid levels, (ii) brain imaging and (iii) cognitive outcomes; this is part of a larger ongoing trial in which the effect of the APOE genotype will be considered. Participants will be followed up for 2 years, with 1 May 2025 as the estimated completion date (https://clinicaltrials.gov/ct2/show/NCT03613844).

## 4. Materials and Methods

### 4.1. Animals

Aged (18–20-month-old) female transgenic Tg2576 mice and their negative littermates were used in this study. Tg2576 mice overexpress the human amyloid precursor protein (hAPP), with the Swedish (K670N/M671L) familial AD mutation under control of the prion promoter [67]. Animals were housed 4–6 per cage and maintained in a temperature-controlled environment on a 12 h light–dark cycle with free access to food and water. All procedures were carried out in accordance with the current European and Spanish regulations (2010/63/EU; RD52/2013) and the study was approved by the Ethical Committee of the University of Navarra.

### 4.2. Treatment

Mice bearing the same genotype were randomly divided in two groups, treatment and control, resulting in four experimental groups of similar dimensions (n = 9–13). DHA was supplemented via daily intragastric administration of 450 mg/kg NuaDHA 1000 (Nua Biological Innovations S.L.Erandio, Vizcaya, Spain) diluted 1:3 *v*/*v* in sesame oil (Sigma-Aldrich, St. Louis, MO, USA), while control animals received the same quantity of the vehicle.

#### 4.2.1. Fear Conditioning Test

The fear conditioning (FC) test paradigm was used to determine whether DHA supplementation improved fear memory. In the habituation phase of this behavioral test, mice were put in the conditioning chamber with no stimuli and allowed to explore and become familiar with the environment for 3 min. After 24 h, animals underwent the training phase: they were placed in the same chamber and received two foot shocks (0.2 mA) of 2 s after 90 and 120 s, respectively. Eventually, after an additional 30 s in the chamber, the mice were returned to their home cages. The following day, the mice were submitted to the actual test consisting of spending 2 min in the conditioning chamber with no stimuli. Freezing behavior during this time, indicating fear memory, was recorded and expressed as a percentage. The behavioral study was carried out during the light phase, from 9:00 to 16:00 h, using a StartFear system (Panlab, Barcelona, Spain).

#### 4.2.2. Morris Water Maze (MWM) Test

The task was carried out in a StartFear system (Panlab), as described by the authors in [68]. Briefly, during the training phase (visible platform, no visible cues) the animals learned to find a platform raised above the surface of the water during three consecutive days with eight trials per day. During the next five consecutive days (four trials per day), mice were trained to locate the platform that was submerged 1 cm beneath the water surface with the help of some visible cues present in the walls of the swimming pool. Finally, 24 h after the last trial on day five, memory retention was tested via a probe trial. All trials were monitored using a camera connected to a SMART-LD program (Panlab) for subsequent analysis of escape latencies during the visible and hidden platform phases, and the percentage of time spent in each quadrant of the pool during the probe trial. All experimental procedures were carried out by personnel who were blind to the different groups.

#### 4.2.3. Determination of Aβ42 Levels

Levels of soluble and insoluble Aβ42 were measured by using a sensitive ELISA kit (Invitrogen, Thermo Fisher Scientific, Inc., Waltham, MA, USA). The prefrontal cortexes of the Tg2576 mice receiving the vehicle (n = 4) or treatment with DHA (n = 6) were homogenized in a buffer containing 10 mM Tris−HCl pH = 7.5, 1 mM NaF, 0.1 mM Na3VO4, 2% SDS and protease inhibitors in order to free the soluble amyloid oligomers. The homogenates were sonicated for 2 min, left on ice for 20 min and centrifuged at 13,000× *g* for 13 min at 6 °C. The presence of insoluble Aβ42 aggregates was evaluated in the parietal cortical tissue (n = 5), homogenized with a 5M guanidine in 50 mM Tris-HCl (pH = 8) buffer. In both cases, the assay was performed according to the manufacturer’s instructions after determining the most suitable dilution, and the resulting data were normalized and represented as Aβ42 content with respect to protein amount (pg/µg).

#### 4.2.4. Dendritic Spine Density Measurement by Golgi–Cox Staining

One hemisphere of each of the brains of n = 3 mice belonging to each group was stained using a modified Golgi–Cox method, allowing the visualization of dendritic spines. The half-brains were soaked in a Golgi–Cox solution (1% potassium dichromate, 1% mercury chloride, 0.8% potassium chromate) for 48 h at RT and protected from light, after which, the solution was then renewed and tissues were left there for a further 3 weeks. Then, brains were washed with distilled water and incubated in 90% ethanol for 30 min before cutting 200 μm thick coronal sections using a vibratome. The slices were incubated in 70% ethanol, washed with distilled water, reduced in 16% ammonia for one hour, fixed in 1% sodium thiosulfate for 7 min and washed again. Once placed on the microscope slides, the preparations were dehydrated using an increasing alcohol graduation and mounted with DPX Mountant (VWR, BDH Prolabo). A Nikon Eclipse E600 light microscope was used to visualize dendritic spines, and Z-stack images were obtained using a digital camera (Nikon DXM 1200 F) at a resolution of 1000–1500 dots per inch (dpi). Spine density was determined in the secondary apical dendrites of CA1 hippocampal pyramidal cells arising at distances from the soma of between 100 and 200 μm, where spine density is considered relatively uniform in CA1 pyramidal neurons [69]. Average data were obtained by quantifying the spine density in 3 neurons of 3 brain slices obtained from each brain (n = 27 dendrites for each group). 

### 4.3. Immunoblotting

For Western blot analysis of APP-derived fragments, protein extracts were mixed with a tricine sample buffer 1:2 (Bio-Rad, Hercules, CA, USA).) and 2% βME; then, they were boiled for 5 min. Proteins were separated in a CriterionTM Tris-Tricine 10–20% gradient precast gel (Bio-Rad, Hercules, CA, USA) and transferred to a PVDF membrane (Hybond LFP, Amersham Biosciences, Little Chalfont, UK) using the Trans-Blot Turbo Transfer System. For Western blot analysis of GFAP and synaptic protein, samples were mixed with a 6X Laemmli sample buffer and resolved onto SDS-polyacrylamide gels and transferred to a nitrocellulose membrane using the Trans-Blot Turbo Transfer System. In all cases, the membranes, blocked with 5% milk in TBS for 1 h at RT, were incubated overnight with the following primary antibodies: mouse monoclonal 6E10 (amino acids 1–16 of Aβ peptide, 1:1000, Covance, San Diego, CA, USA), rabbit monoclonal anti-GFAP (1:1000, Sigma-Aldrich, St. Louis, MO, USA), mouse monoclonal anti-actin (1:100,000, Sigma-Aldrich, St. Louis, MO, USA). 

Three washes in TBS/Tween-20 were performed prior to 1 h incubation with HRP-conjugated anti-rabbit or anti-mouse antibody (1:5000, Santa Cruz Biotechnology, Santa Cruz, CA, USA). Antibody binding to specific bands was visualized via an enhanced chemiluminescence system (ECL, GE Healthcare Bioscience, Uppsala, Sweden or Pierce™ ECL Plus, Thermo Scientific™). Images were acquired and quantified using Image Lab™ Software (Bio-Rad).

#### 4.3.1. RNA-Seq

RNA was extracted from the hippocampal tissue using Trizol Reagent (Sigma-Aldrich) and its integrity was confirmed via Agilent RNA Nano LabChips (Agilent Technologies, Santa Clara, CA, USA). A total of 1 μg of the total RNA was used to construct cDNA libraries with the TruSeq Stranded mRNA Kit (Illumina, San Diego, CA, USA), as described by Perez-Gonzalez et al., 2021 [68] and RNA sequencing data analysis was performed using the same workflow [69]. Genes were selected as differentially expressed using a *p*-value cut-off of *p* < 0.01. Further functional and clustering analyses and graphical representations were performed using R/Bioconductor [70] and QIAGEN IPA [71]. This software calculates significance values (*p*-values) between each biological or molecular event and the imported molecules based on the Fisher’s exact test (*p ≤* 0.05). The IPA comparison analysis considers and hierarchically reports the signaling pathway rank according to the calculated *p*-value.

#### 4.3.2. Quantitative Real-Time PCR

The RNA was treated with DNase at 37 ºC for 30 min and reverse-transcribed into cDNA using SuperScript^®^ III Reverse Transcriptase (Invitrogen). Quantitative real-time PCR was performed to quantify Nr4a2 expression, as described by Perez-Gonzalez et al., 2021 [68]. Assays were carried out in triplicate using the Power SYBR Green PCR Master Mix (Applied Biosystems, Warrington, UK) and the corresponding specific primers for Nr4a2 (Fw: 5′ 3′, Rev: 5′ 3′) and for the internal control 36B4 (Fw: 5′AACATCTCCCCCTTCTCCTT 3′, Rev: 5′ GAAGGCCTTGACCTTTTCAG 3′). 

### 4.4. Statistical Analysis

The results were processed for statistical analysis using GraphPad PRISM, version 5.03. Unless otherwise indicated, results are presented as the mean ± standard error of the mean (SEM). Normal distribution of the data was checked via the Shapiro–Wilk test. Two-way analysis of variance (ANOVA) and Tukey’s post hoc test were used for statistical analyses of the data. In the MWM test, latencies to find the platform were analyzed by two-way repeated measures ANOVA test (genotype × trial) followed by the Bonferroni’s post hoc test to compare the cognitive status among groups. Statistical significance was set at * *p* ≤ 0.05, ** *p* ≤ 0.01 or *** *p* ≤ 0.001.

## 5. Conclusions

In conclusion, our study serves to shed light on the important effect of polyunsaturated fatty acids, particularly DHA, on brain function. Essential neuronal functions related to synaptogenesis, neuritogenesis, the branching of neurites, the density of dendritic spines and the outgrowth of axons, among others, were upregulated upon DHA treatment, thus suggesting that DHA may be considered as an efficient candidate to maintain, protect or even restore synaptic plasticity in neurological disorders that cause synaptic deficits.

## Figures and Tables

**Figure 1 pharmaceutics-15-00082-f001:**
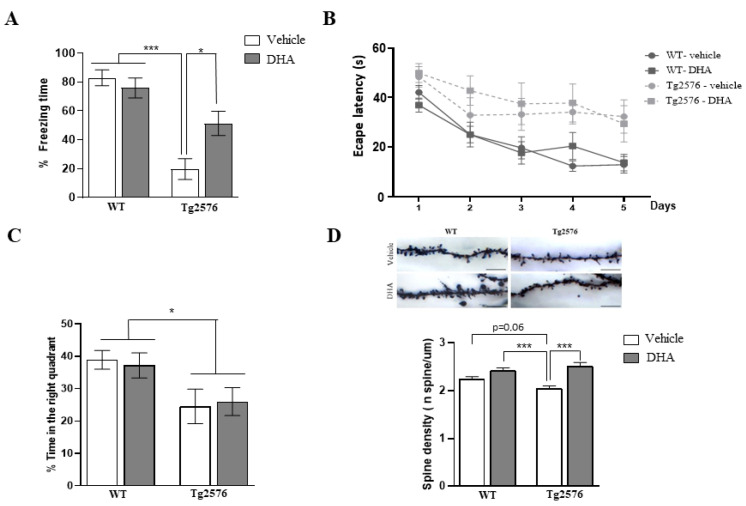
Wild-type (WT) and Tg2576 mice received daily DHA or vehicle and were analyzed 5 weeks later using fear conditioning and the Morris water maze (MWM). (**A**) Freezing behavior from WT and Tg2576 mice treated with DHA or vehicle. Data represent the percentage of freezing time during the test. (**B**) Escape latency of the hidden platform in the MWM test for the WT and Tg2576 mice treated with DHA or vehicle (**C**) Percentage of time spent in correct quadrant during the probe test at day 6. (**D**) Dendritic spine density in CA1 hippocampal neurons from WT and Tg2576 mice treated with DHA or vehicle. In all figures, results are expressed as mean ± SEM (n = 8–11 per group). Two-way analysis of variance (ANOVA) and Tukey’s post hoc test were used for statistical analyses, * *p* ≤ 0.05, *** *p* ≤ 0.001.

**Figure 2 pharmaceutics-15-00082-f002:**
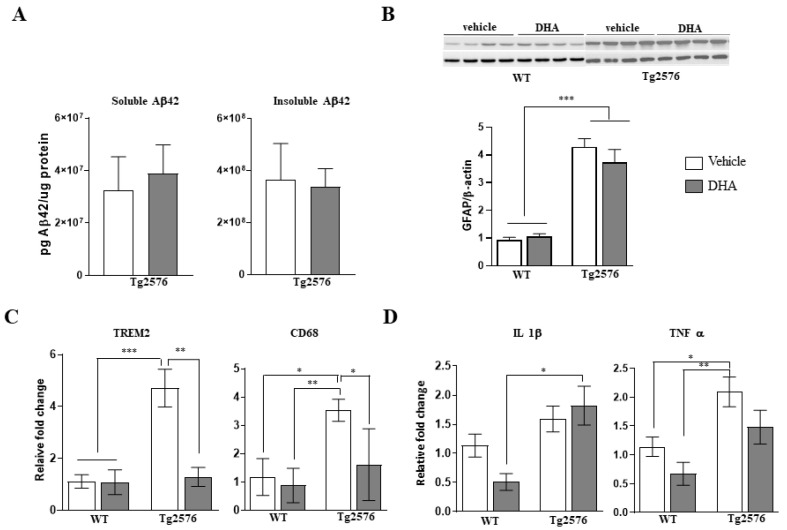
Amyloid pathology and neuroinflammatory markers in the brains of WT and Tg2576 mice chronically treated with DHA. (**A**) Levels of soluble and insoluble human Aβ, as quantified by ELISA in Tg2576 mice treated with DHA or vehicle. (**B**) Western blots showing the effects of DHA or vehicle treatment in WT and Tg2576 mice on GFAP expression, normalized to actin, in 2% SDS hippocampal extracts. (**C**,**D**) The mRNA expression level of microglia markers (TREM2 and CD68) and proinflammatory markers (IL1β and TNFα) were assessed by qRT-PCR. In all figures, results are expressed as mean ± SEM (n = 4–8 per group). Two-way analysis of variance (ANOVA) and Tukey’s post hoc test were used for statistical analyses, * *p* ≤ 0.05, ** *p* ≤ 0.01, *** *p* ≤ 0.001.

**Figure 3 pharmaceutics-15-00082-f003:**
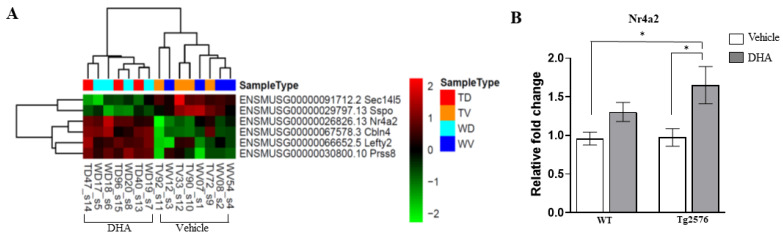
Common genes differentially expressed between vehicle and DHA treatment in both wild-type and Tg2576 mice. (**A**) Differentially expressed genes clusters upon DHA (D) treatment in wild-type (W) and Tg2576 (T) mice. (**B**) The mRNA expression level of Nr4a2 was assessed by qRT-PCR. Results are expressed as mean ± SEM (n = 4–8 per group). Two-way analysis of variance (ANOVA) and Tukey’s post hoc test was used for statistical analyses, * *p* ≤ 0.05.

**Figure 4 pharmaceutics-15-00082-f004:**
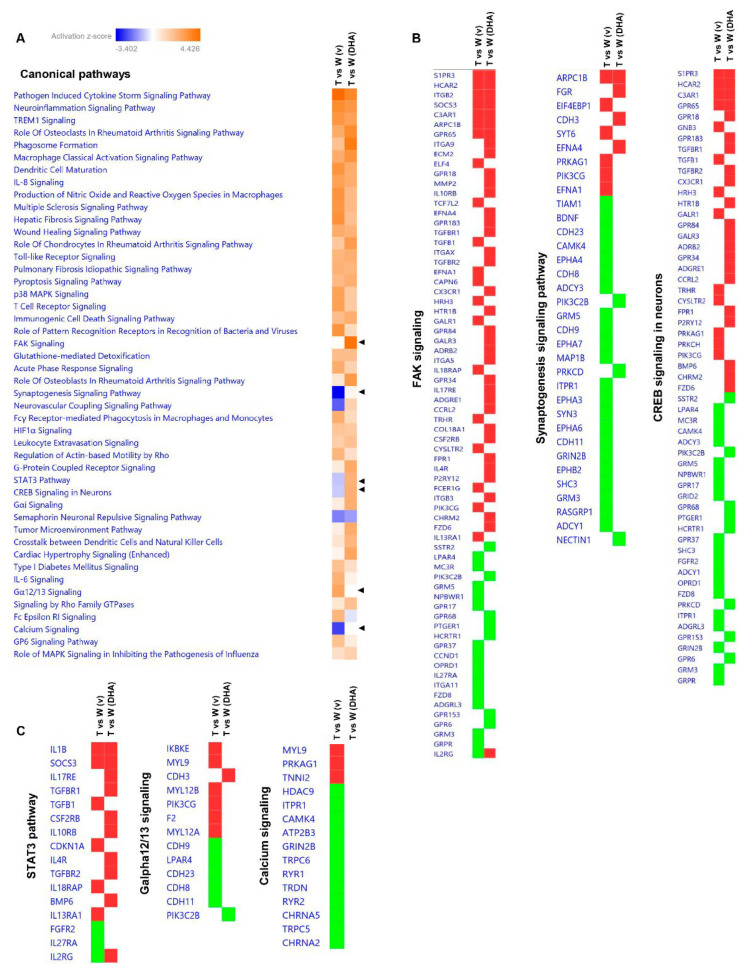
Predictive activation profile of pathways upon DHA treatment. Based on transcriptomic datasets, systems biology analyses were performed using IPA software to obtain the activation prediction of significantly altered pathways (**A**). The activation z-score was calculated as previously described with IPA. Blue and orange squares indicate inhibition and activation directionality, respectively. Black triangles refer to specific processes with an activation score associated with DHA treatment (**B**,**C**). T: Tg2576, W: wildtype, v: vehicle. Red: up-regulation; green: down-regulation.

**Figure 5 pharmaceutics-15-00082-f005:**
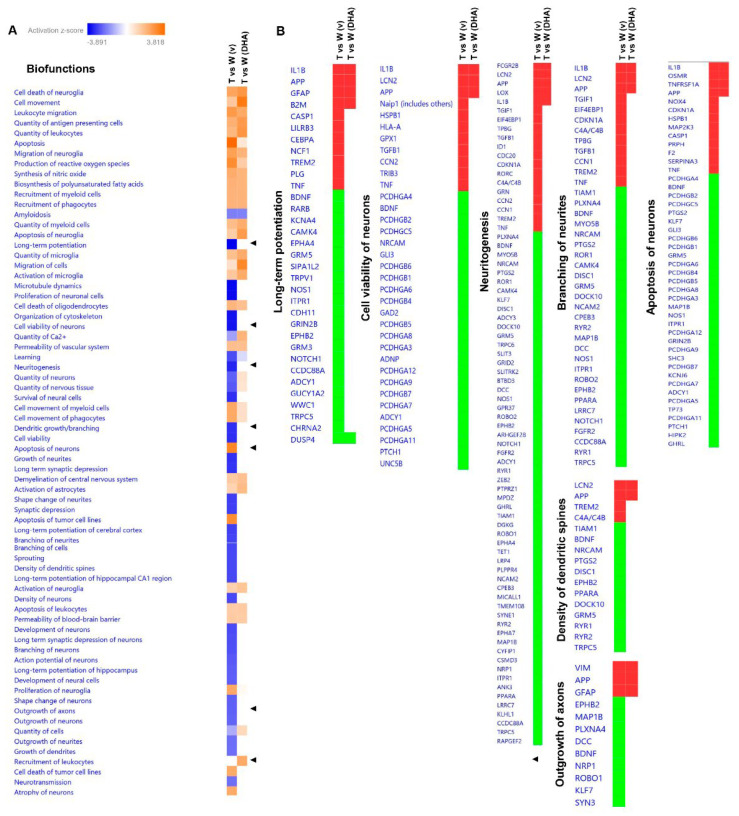
Predictive activation profile of neuronal functions upon DHA treatment. Systems biology analyses were performed using IPA software to generate the activation prediction of significantly altered neuronal functions (**A**). Blue and orange squares indicate inhibition and activation directionality, respectively. Black triangles refer to specific functions whose restoration is associated to DHA treatment (**B**). T: Tg2576, W: wildtype, v: vehicle. Red: up-regulation; green: down-regulation.

**Figure 6 pharmaceutics-15-00082-f006:**
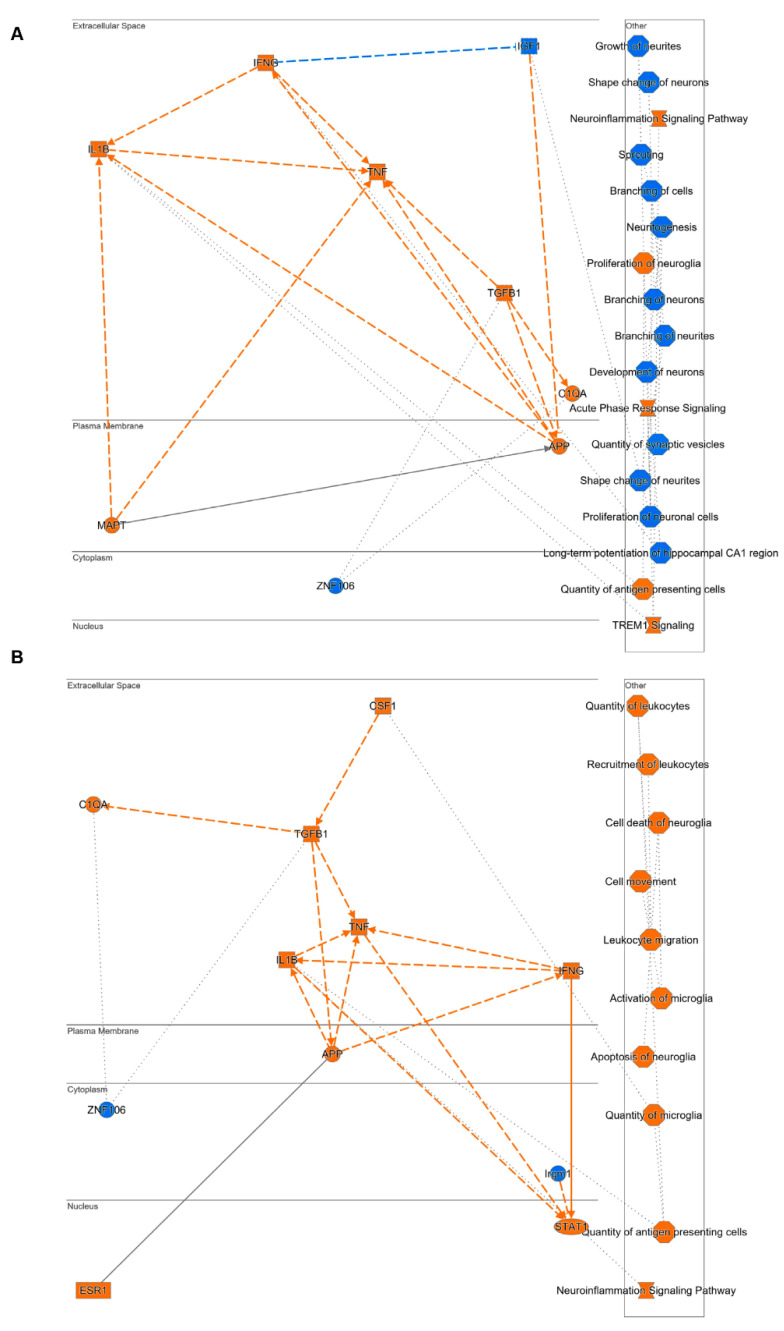
Generation of mechanistic hypothesis based on differential transcriptomic profiling obtained in Tg2576-vehicle vs. wild type-vehicle (**A**) and Tg2576-DHA vs. wild type-DHA (**B**) groups. The functional networks are based on a precomputed table containing inferred relationships between molecules, functions, diseases, and pathways obtained and scored by a machine learning algorithm operating entirely on prior knowledge. The heuristic graph algorithm present in IPA software was optimized to create a manageable network that brings together the most significantly activated (in orange; positive z-score) or inhibited (in blue; negative z-score) upstream regulators, diseases, functions and pathways from the differential RNA-seq datasets.

## Data Availability

RNA-seq analysis data are available at NCBI Gene Expression Omnibus (GEO) under accession number GSE217430. Accessed on 7 November 2022. For information on GEO linking please refer to: https://www.ncbi.nlm.nih.gov/geo/info/linking.html, (accessed on 7 November 2022).

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
