# Peer review of "Docosahexaenoic Acid Ameliorates Contextual Fear Memory Deficits in the Tg2576 Alzheimer’s Disease Mouse Model: Cellular and Molecular Correlates"

_pharmaceutics, 2022, doi:10.3390/pharmaceutics15010082_

Round 1

Reviewer 1 Report

The study deals with the effect of DHA supplementation via oral gavage on cognitive tests, beta-amyloid, hippocampal CA1 spine density and transcription of genes related to neuronal development and function in Tg2576 in comparison to wt mice. The study apparently includes a large amount of data, however most of them regard gene expression levels and need much more functional data to attract consideration. On the other side, the only appreciable effect of DHA supplementation in Tg2576 mice regards an apparent improvement of contextual memory in the fear conditioning test, which however is not confirmed in the Morris water navigation task. In summary, results add very few if any to the topic.

Abstract: please change “…and, therefore, DHA could be an effective diet supplement for AD-dementia prevention” to “and, therefore, DHA should be tested in humans as diet supplement for AD-dementia prevention”. Or similar.

Results, Figure 2D: spine density in CA1 in Tg2576 mice is similar to WT animals. This is quite unexpected (see e.g. https://www.jneurosci.org/content/31/10/3926). In any case, such an important result of DHA treatment should be documented first of all through microphotographs.

Methods: the Morris water navigation task is not described.

In general, the manuscript needs careful correction for a lot of typos, grammar and formatting mistakes.

Author Response

Reviewer #1 (Comments to the Author):

  1. The study deals with the effect of DHA supplementation via oral gavage on cognitive tests, beta-amyloid, hippocampal CA1 spine density and transcription of genes related to neuronal development and function in Tg2576 in comparison to wt mice. The study apparently includes a large amount of data, however most of them regard gene expression levels and need much more functional data to attract consideration. On the other side, the only appreciable effect of DHA supplementation in Tg2576 mice regards an apparent improvement of contextual memory in the fear conditioning test, which however is not confirmed in the Morris water navigation task. In summary, results add very few if any to the topic.

 Thank you very much for the comments. We have further discussed the "apparent discrepancy" in the results of the two memory tests performed. Furthermore, we believed that this study adds important information regarding the effect of DHA on cognitive function. Different mechanisms have been proposed trying to explain a possible beneficial effect and, with our transcriptomic analysis revealed that the effect is through increasing synapse formation rather than by decreasing amyloid pathology.  It is important to consider that in the human condition synapse loss precedes neuronal death and represents the prodromal phase of the disease, which might be represent by AD mice with an established AD-phenotype as the ones of our study (18-month-Tg2576 mice). Accordingly, it seems that beneficial effects of DHA, as was already mention, might not be sufficient to restore memory deficits in people with an already established pathology but may be effective in preventing or delaying disease progression towards dementia mild cognitive impairment patients or in an early stage of the AD.  Thus, our results suggest that DHA could be an effective diet supplement for AD-dementia prevention.

  1. Abstract: please change “…and, therefore, DHA could be an effective diet supplement for AD-dementia prevention” to “and, therefore, DHA should be tested in humans as diet supplement for AD-dementia prevention”. Or similar.

 Thank you very much for you comment, we have already change it.

  1. Results, Figure 2D: spine density in CA1 in Tg2576 mice is similar to WT animals. This is quite unexpected (see e.g. https://www.jneurosci.org/content/31/10/3926). In any case, such an important result of DHA treatment should be documented first of all through microphotographs.

Thank you very much for you comment, we have included representative images of the dendrites whre synapse density have been measure and we have discussed the lack of significance (although there is a clear trend) between dendritic spine density in Tg2576 and wild type mice (see discussion section of the new version of the manuscript).

  1. Methods: the Morris water navigation task is not described.

Sorry for the mistake; the Morris has been described in the new version of the manuscript.

  1. In general, the manuscript needs careful correction for a lot of typos, grammar and formatting mistakes.

We have sent the manuscript for editing.

Reviewer 2 Report

The study by Badesso et al "Docosahexaenoic Acid ameliorates memory deficits in the 2 Tg2576 Alzheimer’s Disease Mouse Model" is an interesting addition to the literature about DHA treatments for dementia. In general, the paper is fine, the major problem is the lack of discussion on the water maze data, where there is no improvement of memory. Similarly, the methods are missing the water maze description.

Author Response

Thank you very much for the comments. We have further discussed the "apparent discrepancy" in the results of the two memory tests performed. We have added the missing Morris water maze in the methodology and we have sent the manuscript for editing. Thus, I hope you will find  the new version of the manuscript suitable for publication.

Round 2

Reviewer 1 Report

The revised version has been improved as regards presentation of methods and results, however the set of data remains basically the same. Therefore even my evaluation remains the same as previously, i.e.: “The study apparently includes a large amount of data, however most of them regard gene expression levels and need much more functional data to attract consideration. On the other side, the only appreciable effect of DHA supplementation in Tg2576 mice regards an apparent improvement of contextual memory in the fear conditioning test, which however is not confirmed in the Morris water navigation task. In summary, results add very few if any to the topic.”

Of course, in general the study has no apparent major flaws, however at least the title should be changed to avoid overrepresentation of actual results, e.g. “Docosahexaenoic Acid ameliorates contextual memory in the fear conditioning test in the Tg2576 Alzheimer’s Disease Mouse Model: cellular and molecular correlates”, or similar.

Author Response

Response to Reviewers

Pharmaceutics manuscript ID: pharmaceutics-2064059: " Docosahexaenoic Acid ameliorates memory deficits in the Tg2576 Alzheimer’s Disease Mouse Model”.

We thank the reviewers for their careful reading and helpful comments. The reviewers found our manuscript to be improved however one of the reviewers still have some comments that we have discussed in this letter

Reviewer #1 (Comments to the Author):

The revised version has been improved as regards presentation of methods and results, however the set of data remains basically the same. Therefore even my evaluation remains the same as previously, i.e.: “The study apparently includes a large amount of data, however most of them regard gene expression levels and need much more functional data to attract consideration. On the other side, the only appreciable effect of DHA supplementation in Tg2576 mice regards an apparent improvement of contextual memory in the fear conditioning test, which however is not confirmed in the Morris water navigation task. In summary, results add very few if any to the topic.”

Of course, in general the study has no apparent major flaws, however at least the title should be changed to avoid overrepresentation of actual results, e.g. “Docosahexaenoic Acid ameliorates contextual memory in the fear conditioning test in the Tg2576 Alzheimer’s Disease Mouse Model: cellular and molecular correlates”, or similar.

Thank you again for your valuable opinion and suggestions. Although I can agree with you that the study is not groundbreaking, I do think that it brings something new and fresh to the field. Following a very controlled high dosage regimen of pure DHA, we have been able to partially reverse the cognitive deficit in an AD mouse model, long after the onset of symptoms. We also provided a transcriptomic analysis to identify the pathways and/or molecules that have been modified by the treatment. Our results revealed an important genotype-dependent effect of DHA, and suggest that a high dose of daily DHA might be beneficial for AD dementia prevention.

Regarding the different effect of DHA in both memory test, as we have already discussed, it confirmed that the memories tested in the fear memory and in the Morris water maze are different. Whereas the contextual fear conditioning is perhaps the simplest form of learning from an operational perspective, and it requires a single training session, the cognitive mapping in spatial tasks such as Morris water maze involve complex interactions between the hippocampus and cortex (among other regions) for making associations between motion and visual information during the early steps of spatial map formation (see reference*). In our hands, the treatment with DHA is able to ameliorate the fear memory but is not able to established all the circuits and connections required to form the spatial memory. We have modified the title of the manuscript according the suggestion raised by the reviewer.

*D’Hooge R, De Deyn PP. Applications of the Morris water maze in the study of learning and memory. Brain Res Rev. 2001;36:60–90.
